# Learning Interpretable Dynamics from Images of a Freely Rotating 3D Rigid Body

## Abstract

In many real-world settings, image observations of freely rotating 3D rigid bodies, such as satellites, may be available when low-dimensional measurements are not. However, the high-dimensionality of image data precludes the use of classical estimation techniques to learn the dynamics and a lack of interpretability reduces the usefulness of standard deep learning methods. In this work, we present a physics-informed neural network model to estimate and predict 3D rotational dynamics from image sequences. We achieve this using a multi-stage prediction pipeline that maps individual images to a latent representation homeomorphic to $\mathbf{SO}(3)$, computes angular velocities from latent pairs, and predicts future latent states using the Hamiltonian equations of motion with a learned representation of the Hamiltonian. We demonstrate the efficacy of our approach on a new rotating rigid-body dataset with sequences of rotating cubes and rectangular prisms with uniform and non-uniform density.

## 1 Introduction

Images of 3D rigid bodies in motion are available across a range of application areas and can give insight into system dynamics. Learning dynamics from images has applications to planning, navigation, prediction, and control of robotic systems. Resident space objects (RSOs) are natural or man-made objects that orbit a planet or moon and are examples of commonly studied, free-rotating rigid bodies. When planning proximity operation missions with RSOs—collecting samples from an asteroid (Williams et al., 2018), servicing a malfunctioning satellite (Flores-Abad et al., 2014), or active space debris removal (Mark and Kamath, 2019)—it is critical to correctly estimate the RSO dynamics in order to avoid mission failure. Space robotic systems typically have access to onboard cameras, which makes learning dynamics from images a compelling approach for vision-based navigation and control.

Previous work Allen-Blanchette et al. (2020); Zhong and Leonard (2020); Toth et al. (2020) has made significant progress in learning dynamics from images of planar rigid bodies. Learning dynamics of 3D rigid-body motion has also been explored with a variety of types of input data Duong and Atanasov (2021); Byravan and Fox (2017); Peretroukhin et al. (2020). Duong and Atanasov (2021) uses state measurement data (i.e. rotation matrix and angular momenta), while Peretroukhin et al. (2020) learn the underlying dynamics in an overparameterized black-box model. The combination of deep learning with physics-based models allows models to learn dynamics from high-dimensional data such as images (Allen-Blanchette et al., 2020; Zhong and Leonard, 2020; Toth et al., 2020). However, as far as we know, our method is the first to use the Hamiltonian formalism to learn 3D rigid-body dynamics from images

Kinematics and dynamics of 3D rigid body rotation are both fundamental to accomplishing the goals of this paper. The kinematics describe the rate of change of rigid body orientation as a function of the orientation and the angular velocity. Our method integrates the kinematic equations to compute the orientation trajectory in latent space using the latent angular velocity. The dynamics describe the rate of change of the angular velocity as a function of the angular velocity and the moment-of-inertia matrix $\mathbf{J}$, which depends on the distribution of mass over the rigid body volume. $\mathbf{J}$ is unknown and cannot be computed from knowledge of the external geometry of the rigid body, except in the special case in which the mass is known and the mass is uniformly distributed over the rigid body volume. In our framework, we learn the dynamics from the motion of the rigid body, not from the external

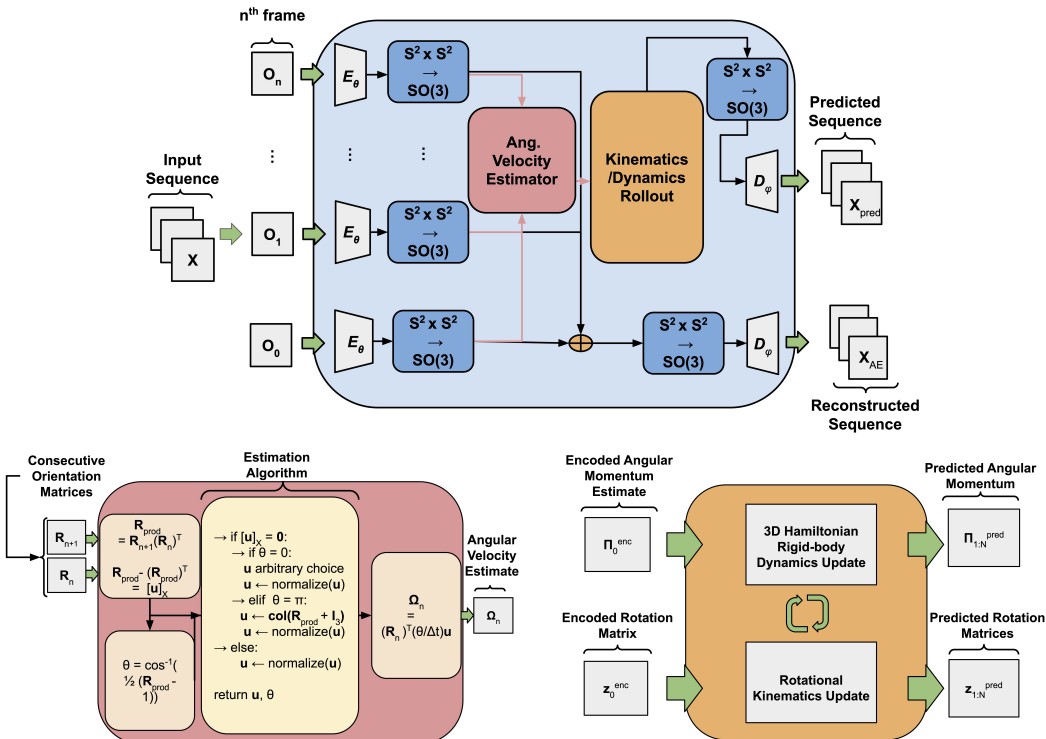

Figure 1: Schematic of model architecture (top). The architecture combines an auto-encoding neural network with a Hamiltonian dynamics model for 3D rigid bodies (bottom-right). The encoder maps a sequence of images to a sequence of latent states in $\mathbf{SO}(3)$. We estimate angular velocity and momentum, then predict future orientation and momentum using the learned Hamiltonian (bottom-left). Each future latent state is decoded into an image using only the predicted orientation.

geometry since the special case is not practical. Importantly, the difference between the dynamics of a uniformly distributed mass and a non-uniformly distributed mass inside the same external geometry is significant. We show that we can learn these very different dynamics even when the external geometry is the same. By integrating the learned dynamics, we can predict future latent angular velocity. Works such as [1,2] estimate the transformation between image pairs, which can then be used to generate image trajectories. However, this approach implicitly assumes the object evolves with a constant velocity, an assumption that is not true in the general case. Our work addresses this limitation by estimating the angular acceleration and mass distribution for the object in addition to its angular velocity.

In this paper we introduce a model, with architecture depicted in Figure 1, that is capable of (1) learning 3D rigid-body dynamics from images, (2) predicting future image sequences in time, and (3) providing a low-dimensional, interpretable representation of the latent state. Our model incorporates the Hamiltonian formulation of the dynamics as an inductive bias to facilitate learning the moment of inertia tensor $\mathbf{J} \in \mathbb{R}^{3\times3}$ and an auto-encoding map between images and $\mathbf{SO}(3)$. The efficacy of our approach is demonstrated through long-term image prediction and its additional latent space interpretability.

## 2 RELATED WORK

### 2.1 HAMILTONIAN AND LAGRANGIAN PRIORS IN LEARNING DYNAMICS

Physics-guided machine learning approaches with Lagrangian and Hamiltonian priors often use state-control (e.g. 1D position and control) trajectories (Ahmadi and Khadir, 2020; Chen et al., 2020; Cranmer et al., 2020a; Duong and Atanasov, 2021; Finzi et al., 2020; Greydanus et al., 2019;

Gupta et al., 2019; Lutter et al., 2018; Zhong et al., 2020b;a) to train their models. Lagrangian-based methods (Lutter et al., 2018; Gupta et al., 2019; Cranmer et al., 2020a; Finzi et al., 2020; Allen-Blanchette et al., 2020; Zhong and Leonard, 2020) parameterize the Lagrangian function $\mathcal{L}(q, \dot{q})$ using neural networks to derive the Euler-Lagrange dynamical equations (ELEs). These methods have been used to learn inverse dynamics models to derive control laws from data (Lutter et al., 2018) and approximate the ELEs with neural networks for forward prediction (Cranmer et al., 2020a; Finzi et al., 2020). Greydanus et al. (2019) create Hamiltonian-based priors by introducing Hamiltonian Neural Networks (HNNs), a method for parameterizing the Hamiltonian, $H$, as a neural network and calculating the symplectic gradients to construct Hamilton's equations. Zhong et al. (2020b) builds upon the ideas of (Greydanus et al., 2019) by further decomposing the Hamiltonian into different coordinate-dependent functions, such as the system's potential energy, inverse mass matrix, and input matrix—providing additional structure. This method was extended to more general systems by Zhong et al. (2020a), where they introduce dissipative forces into their formulation. These models focus on using a dataset of state-control trajectories whereas our model uses image sequences. Moreover, while (Zhong et al., 2020b;a) have elements of their work that address rigid-body systems, these models focus on planar systems and do not use images.

## 2.2 Using images for learning dynamics

Image-based learning models focus on using a set of images that represent the state trajectories for systems as data to train their models for a variety of tasks such as dimensionality reduction and learning dynamics. Approaches that model the dynamics of the system (Li et al., 2020; Zhong and Leonard, 2020; Allen-Blanchette et al., 2020; Toth et al., 2020) learn dynamics from image-state data but only for either 2D planar systems or systems with dynamics in $\mathbb{R}^n$, using pixel images. Zhong and Leonard (2020) use a Lagrangian prior and a coordinate-aware encoder and decoder in their variational auto-encoding neural network to estimate the underlying dynamics and do video prediction with control. Allen-Blanchette et al. (2020) focus on learning dynamics from images for similar systems but using a non-variational approach, encouraging an interpretable latent space for long-term video prediction. Most similar to our work, Toth et al. (2020) use a Hamiltonian-based approach to learning dynamics from images. The work of Toth et al. differs from ours in that we do not use a variational auto-encoding neural network, our model is used for systems with configuration space on $\mathbf{SO}(3)$, and our work focuses on learning a physically interpretable latent space while Toth et al. have a much higher dimension latent space than the physical configuration space.

## 2.3 Learning dynamics for rigid bodies

Duong and Atanasov (2021) investigate learning rigid-body dynamics from state-control trajectory data using a Hamiltonian approach. In their work, Duong and Atanasov focus on systems with configuration spaces on the special orthogonal groups and special Euclidean groups (e.g. $\mathbf{SO}(2)$, $\mathbf{SO}(3)$, and $\mathbf{SE}(3)$). They parameterize the inverse mass properties, potential energy, and control input functions, which are estimated with neural networks similar to our method. Duong and Atanasov go on to use the estimated pose dynamics to develop control methodologies for stabilization and tracking using the learned model of a quad-rotor system. In contrast, our work focuses on learning the dynamics of a free-rotating rigid body with configuration space on $\mathbf{SO}(3)$ using images.

## 3 Background

### 3.1 The $\mathcal{S}^2 \times \mathcal{S}^2$ parameterization of 3D rotation group

The $\mathcal{S}^2 \times \mathcal{S}^2$ parameterization of the 3D rotation group is a surjective and differentiable mapping with a continuous right inverse Falorsi et al. (2018). Define the $n$-sphere: $\mathcal{S}^n = \left\{ \mathbf{v} \in \mathbb{R}^{(n+1)} \mid v_1^2 + v_2^2 + \ldots + v_{n+1}^2 = 1 \right\}$, and the 3D rotation group: $\mathbf{SO}(3) = \left\{ \mathbf{R} \in \mathbb{R}^{3 \times 3} \mid \mathbf{R}^T \mathbf{R} = \mathbf{I}_3, \ \det(\mathbf{R}) = +1 \right\}$, where $\mathbf{I}_3$ denotes the $3 \times 3$ identity matrix. The $\mathcal{S}^2 \times \mathcal{S}^2$ parameterization of $\mathbf{SO}(3)$ is given by $(u, v) \mapsto (w_1, w_2, w_3)$ where $w_1 = u$, $w_2 = v - v\langle u, v \rangle$, $w_3 = w_1 \times w_2$, where $w_i$ are renormalized to have unit norm. Intuitively, this mapping constructs an orthonormal frame from the unit vectors $u$ and $v$ by Gram-Schmidt orthogonalization. The right inverse of the parameterization is given by $(w_1, w_2, w_3) \mapsto (w_1, w_2)$. Other parameterizations of $\mathbf{SO}(3)$ such as the exponential map ($\mathfrak{so}(3) \mapsto \mathbf{SO}(3)$) and the quaternion map ($\mathcal{S}^3 \mapsto \mathbf{SO}(3)$) do not have continuous inverses and

therefore are more difficult to use in deep manifold regression (Falorsi et al., 2018; Levinson et al., 2020; Brégier, 2021).

## 3.2 3D ROTATING RIGID-BODY KINEMATICS

A rotating 3D rigid body's orientation $\mathbf{R}(t) \in \mathbf{SO}(3)$ changing over time $t$ can be computed from angular velocity $\mathbf{\Omega}(t)$ using the kinematic equations given by the time-rate-of-change of $\mathbf{R}(t)$ as

$$\frac{d\mathbf{R}(\mathbf{t})}{dt} = \mathbf{R}(t)\mathbf{\Omega}_\times(t), \tag{1}$$

where $\mathbf{\Omega}_\times$ is a 3-dimensional skew-symmetric matrix defined by $(\mathbf{\Omega}_\times)\boldsymbol{y} = \mathbf{\Omega} \times \boldsymbol{y}$, with $\boldsymbol{y} \in \mathbb{R}^3$ and $\times$ is the vector cross-product. For computational purposes, 3D rigid-body rotational kinematics are commonly expressed in terms of the quaternion representation $\mathbf{q}(t) \in \mathcal{S}^3$ of the rigid-body orientation $\mathbf{R}(t)$. The kinematics equation 1 written in terms of quaternions (Andrle and Crassidis, 2013) are

$$\frac{d\mathbf{q}(t)}{dt} = \mathbf{Q}(\mathbf{\Omega}(t))\mathbf{q}(t), \quad \mathbf{Q}(\mathbf{\Omega}) = \begin{pmatrix} -\mathbf{\Omega}_\times & \mathbf{\Omega} \\ -\mathbf{\Omega}^T & 0 \end{pmatrix}. \tag{2}$$

## 3.3 3D RIGID-BODY DYNAMICS IN HAMILTONIAN FORM

The canonical Hamiltonian formulation derives the equations of motion for a mechanical system using only the symplectic form and a Hamiltonian function, which maps the state of the system to its total (kinetic plus potential) energy Goldstein et al. (2002). This formulation has been used by several authors to learn unknown dynamics: the Hamiltonian structure (canonical symplectic form) is used as a physics prior and the unknown dynamics are uncovered by learning the Hamiltonian (Greydanus et al., 2019; Zhong et al., 2020b; Toth et al., 2020). Consider a system with configuration space $\mathbb{R}^n$ and a choice of $n$ generalized coordinates that represent configuration. Let $\mathbf{z}(t) \in \mathbb{R}^{2n}$ represent the vector of $n$ generalized coordinates and their $n$ conjugate momenta at time $t$. Define the Hamiltonian function $H : \mathbb{R}^{2n} \mapsto \mathbb{R}$ such that $H(\mathbf{z})$ is the sum of the kinetic plus potential energy. Then the equations of motion derive as

$$\frac{d\mathbf{z}}{dt} = \Lambda_{\mathrm{can}}\nabla_{\mathbf{z}}H(\mathbf{z}), \quad \Lambda_{\mathrm{can}} = \begin{pmatrix} \mathbf{0}_n & \mathbf{I}_n \\ -\mathbf{I}_n & \mathbf{0}_n \end{pmatrix} \tag{3}$$

where $\mathbf{0}_n \in \mathbb{R}^{n \times n}$ is the matrix of all zeros and $\Lambda_{\mathrm{can}}$ is the matrix representation of the canonical symplectic form Goldstein et al. (2002). The equations of motion for a freely rotating 3D rigid body evolve on $T^*\mathbf{SO}(3)$ and describe the evolution of $\mathbf{R}$ given by equation 1 and the evolution of the angular momentum $\mathbf{\Pi}$ given by Euler's equations Goldstein et al. (2002):

$$\frac{d\mathbf{\Pi}}{dt} = \mathbf{\Pi} \times \mathbf{J}^{-1}\mathbf{\Pi}, \tag{4}$$

where $\mathbf{J}$ denotes the moment-of-inertia matrix and $\mathbf{\Pi} = \mathbf{J}\mathbf{\Omega}$. Importantly, the dynamics equation 4 do not depend on $\mathbf{R}$. This is due to the rotational symmetry of a freely rotating 3D rigid body, i.e., it is arbitrary how we assign an inertial frame. Due to this symmetry, the Hamiltonian formulation can be reduced using the Lie-Poisson Reduction Theorem Marsden and Ratiu (1999) to describe the time evolution of $\mathbf{\Pi}$ on $\mathbb{R}^3 \sim \mathfrak{so}^*(3)$, the Lie coalgebra of $\mathbf{SO}(3)$, independent of equation 1. The reduced Hamiltonian $h : \mathfrak{so}^*(3) \mapsto \mathbb{R}$ for the freely rotating 3D rigid body is its kinetic energy:

$$h(\mathbf{\Pi}) = \frac{1}{2}\mathbf{\Pi} \cdot \mathbf{J}^{-1}\mathbf{\Pi}. \tag{5}$$

The reduced Hamiltonian formulation Marsden and Ratiu (1999) is

$$\frac{d\mathbf{\Pi}}{dt} = \Lambda_{\mathfrak{so}^*(3)}(\mathbf{\Pi})\nabla_{\mathbf{\Pi}}h(\mathbf{\Pi}), \quad \Lambda_{\mathfrak{so}^*(3)}(\mathbf{\Pi}) = \mathbf{\Pi}_\times, \tag{6}$$

which can be seen to be equivalent to equation 4. The equations equation 6, called Lie-Poisson equations, generalize the canonical Hamiltonian formulation. The generalization allows for different symplectic forms, i.e., $\Lambda_{\mathfrak{so}^*(3)}$ instead of $\Lambda_{\mathrm{can}}$ in this case, each of which is only related to the latent space and symmetry. Our physics prior is the generalized symplectic form and learning the unknown dynamics means learning the reduced Hamiltonian. This is a generalization of the existing literature

where dynamics of canonical Hamiltonian systems are learned with the canonical symplectic form as the physics prior. (Greydanus et al., 2019; Cranmer et al., 2020a; Chen et al., 2020; Toth et al., 2020). Using the generalized Hamiltonian formulation extends the approach to a much larger class of systems than those described by Hamilton's canonical equations, including rotating and translating 3D rigid bodies, rigid bodies in a gravitational field, multi-body systems, and more.

# 4 LEARNING HAMILTONIAN DYNAMICS ON $\mathbf{SO}(3)$

In this section we outline our approach for learning and predicting rigid-body dynamics from image sequences. The multi-stage prediction pipeline maps individual images to an $\mathbf{SO}(3)$ latent space where angular velocities are computed from latent pairs. Future latent states are computed using the generalized Hamiltonian equations of motion and a learned representation of the reduced Hamiltonian. Finally, the predicted latent representations are mapped to images giving a predicted image sequence.

## 4.1 NOTATION

$N$ denotes the number of image sequences in the dataset and $T$ the length of each image sequence. Image sequences are written $\mathbf{I}_k = \{I_1^k, \ldots, I_T^k\}$ with $I_t^k \in \mathcal{I}$, embedded sequences are written $Z_k = \{z_1^k, \ldots, z_T^k\}$ with $z_t^k \in \mathcal{Z}$, $\mathbf{SO}(3)$ latent sequences are written $\mathbf{R}_k = \{R_1^k, \ldots, R_T^k\}$ with $R_t^k \in \mathbf{SO}(3)$, and quaternion sequences are written $\mathbf{q}_k = \{q_1^k, \ldots, q_T^k\}$ with $q_t^k \in \mathcal{S}(3)$. Quantities generated with the learned dynamics are denoted with a hat (e.g., $\hat{q}$). For all sequences $k \in \{1, \ldots, N\}$.

## 4.2 EMBEDDING TO AN $\mathbf{SO}(3)$ LATENT SPACE

We embed image observations of a rotating rigid body to an $\mathbf{SO}(3)$ latent space using the composition of functions $f \circ \pi \circ E_\theta : \mathcal{I} \mapsto \mathbf{SO}(3)$. The encoding network $E_\theta : \mathcal{I} \mapsto \mathbb{R}^6$ is learned during training, the projection $\pi : \mathbb{R}^6 \mapsto \mathcal{S}^2 \times \mathcal{S}^2$ is defined as $\pi(z) = (u/\|u\|,\ v/\|v\|)$, $u, v \in \mathbb{R}^3$ where $z = (u,\ v)$, and the function $f : \mathcal{S}^2 \times \mathcal{S}^2 \mapsto \mathbf{SO}(3)$ denotes the surjective and differentiable $\mathcal{S}^2 \times \mathcal{S}^2$ parameterization of $\mathbf{SO}(3)$ (see Section 3.1) which constrains embedded representations to the $\mathbf{SO}(3)$ manifold where we compute dynamics.

## 4.3 HAMILTONIAN DYNAMICS ON $\mathbf{SO}(3)$

We predict future $\mathbf{SO}(3)$ latent states using the equations of motion for a freely rotating 3D rigid body (see equations 2 and 6) and the learned moment of inertial tensor $\mathbf{J}_\psi$. We construct the initial state $x_0^k = (q_0^k, \Pi_0^k)$ using the pair of sequential $\mathbf{SO}(3)$ latent states $(R_0^k, R_1^k)$ (see Section 4.2). The quaternion $q_0^k$ is computed using the implementation of a modified Shepperd's algorithm(Markley, 2008) proposed in (Falorsi et al., 2018) and the angular momentum $\Pi_0^k$ is computed as $\Pi_0^k = \mathbf{J}_\psi \Omega_0^k$ where the angular velocity $\Omega_0^k$ is approximated using the algorithm proposed in (Barfoot, 2017) (see also Figure 1: Angular Velocity Estimator). The kinematic equations equation 2 are integrated forward using a Runge-Kutta fourth-order solver (RK45) and a normalization step (Andrle and Crassidis, 2013) to ensure the quaternions are valid.

We decode a predicted image sequence from the predicted quaternion sequence $\mathbf{q}_k$ in three steps. We first transform each $q_t^k$ to $R_t^k$, then apply the right inverse of the $\mathcal{S}^2 \times \mathcal{S}^2$ parameterization of $\mathbf{SO}(3)$, and finally, decode from $\mathcal{S}^2 \times \mathcal{S}^2$ with the decoding neural network $D_\phi$ which is learned during training.

## 4.4 LOSS FUNCTIONS

In this section we describe each component of our loss function: the auto-encoder reconstruction loss $\mathcal{L}_{\text{ae}}$, dynamics reconstruction loss $\mathcal{L}_{\text{dyn}}$, and latent losses functions $\mathcal{L}_{\text{latent, R}}$, $\mathcal{L}_{\text{latent, }\Pi}$, $\mathcal{L}_{\text{energy}}$. The function $\mathcal{L}_{\text{ae}}$ ensures the embedding to $\mathbf{SO}(3)$ is sufficiently expressive to represent the image state, and $\mathcal{L}_{\text{dyn}}$ ensures the dynamics consistency between decoded latent states images and their decoded predicted states. Both $\mathcal{L}_{\text{latent, R}}$ and $\mathcal{L}_{\text{latent, }\Pi}$ ensure the dynamics consistency of encoded images and their predicted states in the latent space. Lastly, $\mathcal{L}_{\text{energy}}$ enforces the conservation of energy between

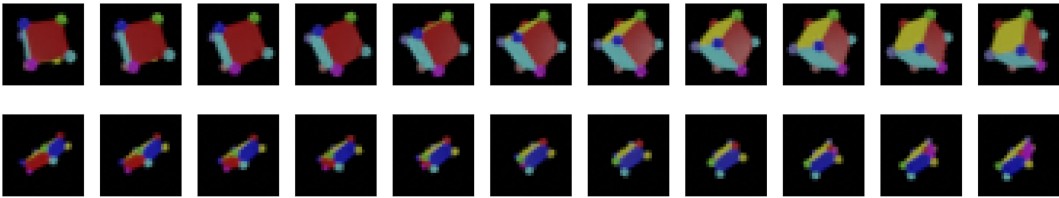

Figure 2: Example training sequences. (Top) Uniform density cube, (bottom) Uniform density prism.

encoded and predicted trajectories. For notational convenience we denote the embedding pipeline $\mathcal{E} : \mathcal{I} \mapsto \mathcal{S}^3$, and the decoding pipeline $\mathcal{D} : \mathcal{S}^3 \mapsto \mathcal{I}$

### 4.4.1 RECONSTRUCTION LOSS

The auto-encoding reconstruction loss is the mean square error (MSE) between the ground-truth image sequence:

$$\mathcal{L}_{\text{ae}} = \frac{1}{NT} \sum_{k=1}^{N} \sum_{t=0}^{T-1} \left\| I_t^k - (\mathcal{D} \circ \mathcal{E})(I_t^k) \right\|_2^2 .$$

The dynamics reconstruction loss function is the MSE between the ground-truth image sequence and the predicted image sequence:

$$\mathcal{L}_{\text{dyn}} = \frac{1}{NT} \sum_{k=1}^{N} \sum_{t=1}^{T} \left\| I_t^k - \mathcal{D}\left(\hat{q}_t^k\right) \right\|_2^2 . \tag{7}$$

### 4.4.2 LATENT LOSS

The latent state loss function is a distance metric on $\mathbf{SO}(3)$ (Huynh, 2009), defined as the MSE between the $3 \times 3$ identity matrix and right-difference between the encoded latent states and the latent states predict using the learned dynamics:

$$\mathcal{L}_{\text{latent, R}} = \frac{1}{NT} \sum_{k=1}^{N} \sum_{t=1}^{T} \left\| \mathbf{I}_3 - R_{\text{enc}_t}^k \left(\hat{R}_t^k\right)^T \right\|_F^2 ,$$

An additional loss is computed over the angular momentum vectors estimated from the encoded latent states (see Figure 1) and the predicted angular momentum vectors using the MSE loss:

$$\mathcal{L}_{\text{latent},\Pi} = \frac{1}{NT} \sum_{k=1}^{N} \sum_{t=1}^{T} \left\| \Pi_{\text{enc}_t}^k - \hat{\Pi}_t^k \right\|_2^2 .$$

### 4.5 ENERGY-BASED LOSS

We encourage conservation of energy in the latent representation using the energy conservation loss:

$$\mathcal{L}_{\text{energy}} = \frac{1}{N(T+1)} \sum_{k=1}^{N} \sum_{t=0}^{T} \left( E_t^k - \bar{E}^k \right)^2 ,$$

$$\bar{E}^k = \frac{1}{T+1} \sum_{t=0}^{T} E_t^k .$$

where, $E_t^k = h(\Pi_{\text{enc}_t}^k ; \mathbf{J}_\psi^{-1})$ is computed using equation equation 5:

## 5 RIGID BODY DATASETS

The lack of exploration of 3D learning tasks creates a deficit of datasets for models designed for 3D dynamics. Previous contributions to the study of learning dynamics from images (Greydanus et al.,

Table 1: Average pixel mean square error over a 30 step unroll on the train and test data on four datasets. All values are multiplied by 1e+3. We evaluate our model and compare to three baseline models: (1) recurrent model (LSTM Hochreiter and Schmidhuber (1997)), (2) NeuralODE (Chen et al. (2018)), (3) HGN (Toth et al. (2020)). Our model outperforms both baseline models in the prediction task across the majority of the datasets. The number parameters for the dynamics models of each baselines are given in the last row of the table.

| Dataset | Ours | | LSTM - baseline | | NeuralODE - baseline | | HGN | |
|---|---|---|---|---|---|---|---|---|
| | TRAIN | TEST | TRAIN | TEST | TRAIN | TEST | TRAIN | TEST |
| Uniform Prism | $\mathbf{2.66 \pm 0.10}$ | $\mathbf{2.71 \pm 0.08}$ | $3.46 \pm 0.59$ | $3.47 \pm 0.61$ | $3.96 \pm 0.68$ | $4.00 \pm 0.68$ | $4.18 \pm 0.0$ | $7.80 \pm 0.30$ |
| Uniform Cube | $\mathbf{3.54 \pm 0.17}$ | $\mathbf{3.97 \pm 0.16}$ | $21.55 \pm 1.98$ | $21.64 \pm 2.12$ | $9.48 \pm 1.19$ | $9.43 \pm 1.20$ | $17.43 \pm 0.00$ | $18.69 \pm 0.12$ |
| Non-uniform Prism | $\mathbf{4.27 \pm 0.18}$ | $6.61 \pm 0.88$ | $4.50 \pm 1.31$ | $\mathbf{4.52 \pm 1.34}$ | $4.67 \pm 0.58$ | $4.75 \pm 0.59$ | $6.16 \pm 0.08$ | $8.33 \pm 0.26$ |
| Non-uniform Cube | $6.24 \pm 0.29$ | $\mathbf{4.85 \pm 0.35}$ | $7.47 \pm 0.51$ | $7.51 \pm 0.50$ | $7.89 \pm 1.50$ | $7.94 \pm 1.59$ | $14.11 \pm 0.13$ | $18.14 \pm 0.36$ |
| Number of Parameters | 6 | | 52400 | | 11400 | | - | |

2019; Toth et al., 2020; Allen-Blanchette et al., 2020; Zhong and Leonard, 2020) have evaluated their models on pixel image sequences of 2D planar dynamics. Their datasets include the pixel pendulum, Acrobot, cart-pole (Zhong and Leonard, 2020), as well 2-body and 3-body problems (Toth et al., 2020). These approaches, which primarily use 2D datasets, are not applicable to our problem of learning 3D rigid-body dynamics. Therefore, we created datasets that demonstrate the rich dynamics behaviors of 3D rotational dynamics through images, capable of being used for 3D dynamics learning tasks. We empirically test the performance of our model on the following datasets:

- **Uniform density cube**: Multi-colored cube of uniform mass density
- **Uniform density prism**: Multi-colored rectangular prism with uniform mass density
- **Non-uniform density cube**: Multi-colored cube with non-uniform mass density
- **Non-uniform density prism**: Multi-colored prism with non-uniform mass density

The uniform mass density cube and prism datasets demonstrate baseline capabilities of our approach while the non-uniform datasets are used to demonstrate capabilities of our model pertinent to applications. The uniform mass density datasets can be learned by estimating a diagonal moment of inertia, since the principle axes of rotation align with our defined body axes. These axes are more intuitive to estimate from a visual perspective. For the uniform cube dataset, inspired by Falorsi et al. (2018), the angular momentum vector is constant. For the uniform prism dataset, there exists a set of initial conditions that result in dynamics that are not locally bounded, and thus more difficult to learn. The non-uniform density datasets are used to demonstrate the capability of our model to learn dynamics when the principle axes of rotation do not align with the set of intuitive body axes of the object. The physical appearance of the objects match those of the uniform mass density datasets; however, the momentum of inertia in the body axes frame will be non-diagonal. This dataset will validate the model's capability to predict changes in mass distribution that may not be visible for failure diagnostics due to broken or shifted internal components.

For each dataset $N = 1000$ trajectories were created. Each trajectory consisted of an initial condition $\mathbf{x}_0 = (\mathbf{R}_0, \mathbf{\Pi}_0)$ pair that was integrated forward in time using a Python-based Runge-Kutta solver for $T = 100$ timesteps with spacing $\Delta t = 10^{-3}$. Initial conditions were chosen such that $(\mathbf{R}_0, \mathbf{\Pi}_0) \sim$ Uniform $(\mathbf{SO}(3) \times S^2)$ with $\mathbf{\Pi}_0$ scaled to have $\|\mathbf{\Pi}_0\|_2 = 50$.

The orientations from the trajectories were passed to Blender Community (2018) to render 28x28 pixel images. Examples of the renderings for both the cube and the rectangular prism can be found in Figure 2. For training, each trajectory is shaped into trajectory windows of sequence length $\tau = 10$, using a sliding window such that each iteration trains on a batch of image sequences of length $\tau$.

## 6 RESULTS

### 6.1 IMAGE PREDICTION

One key contribution of this work is image prediction for freely rotating rigid bodies using the learned dynamics model. The model is capable of high-accuracy future prediction across four datasets. Figure 3 shows the model's performance on the datasets for both short and long-term predictions. The model's performance on the two uniform density datasets (top) is indicative of its capabilities to

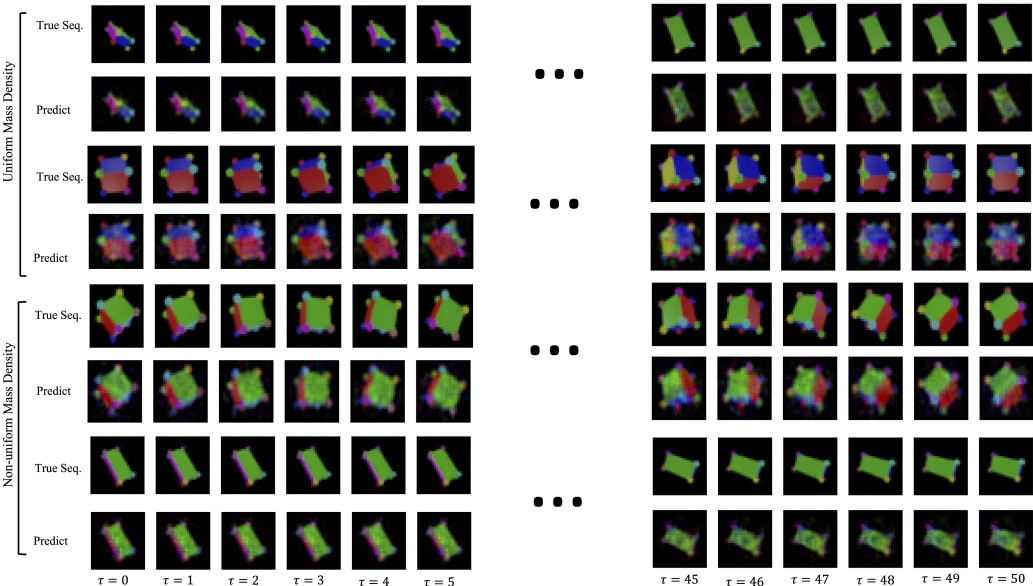

Figure 3: Predicted sequences for uniform and non-uniform mass density datasets given by the model. At prediction time, the model takes the first two images of a sequence, encodes them into our latent space to estimate the angular momentum. The model then predicts and decodes the predicted future states into images of rotating rigid bodies. The prediction results show that the model is qualitatively capable of predicting into the future using images.

predict dynamics and map them to image space. The model's ability to accurately predict future image states for the non-uniform density datasets is also interesting in that it demonstrates that the model is able to predict states accurately even when the density is visually ambiguous. This is particularly important because mass density (and thus rotational dynamics) is not something that can easily be inferred directly from images. The model is compared to three baseline models: (1) an LSTM-baseline, (2) a Neural ODE (Chen et al., 2018)-baseline, and (3) the HGN (Toth et al., 2020) model. Recurrent neural networks like the LSTM-baseline provide a discrete dynamics model. Neural ODE can be combined with a multi-layer perceptron to model and predict continuous dynamics. HGN is a variational model with a Hamiltonian inductive bias. Architecture and training details for each baseline is given in the appendix B. The prediction performance of our model and baselines is shown in Table 1. Our model outperforms the baseline models on most of the datasets with a more interpretable latent space, continuous dynamics, and fewer model parameters –motivating principles for this work.

## 6.2 LATENT SPACE ANALYSIS AND INTERPRETABILITY

Another contribution of this work is the interpretability of the latent space generated by our model from the image datasets. Black-box models with high-dimensional latent states make it very difficult to interpret and inspect the behavior of the latent space. Because our approach encodes all images into a latent space homeomorphic to $\mathbf{SO}(3)$, we have convenient ways to interpret this low-dimensional space.

The form of our latent space provides us a way of inspecting the behavior of our model that previous works lack. In this sense, our approach provides a step towards producing new frameworks for interpreting deep learning models, analyzing failure modes, and using control for dynamic systems with significant structure from prior knowledge.

## 7 CONCLUSIONS

### 7.1 SUMMARY

In this work, we have presented the first physics-informed deep learning framework for predicting image sequences of 3D rigid-bodies by embedding the images as measurements in the configuration space $\mathbf{SO}(3)$ and propagating the Hamiltonian dynamics forward in time. We have evaluated our approach on a new dataset of free-rotating 3D bodies with different inertial properties, and have demonstrated the ability to perform long-term image predictions.

By enforcing the representation of the latent space to be the correct manifold, this work provides the advantage of interpretability over black-box physics-informed approaches. The extra interpretability of our approach is a step towards placing additional trust into sophisticated deep learning models. This work provides a natural path to investigating how to incorporate—and evaluate the effect of—classical model-based control directly to trajectories in the latent space. This interpretability is essential to deploying ML algorithms in safety-critical environments.

### 7.2 LIMITATIONS

While this approach has shown significant promise, it is important to highlight that this has only been tested in an idealized setting. Future work can examine the effect of dynamic scenes with variable backgrounds, lighting conditions, object geometries, and multiple bodies. Perhaps more limiting, this approach currently relies on the ability to train the model for each system being examined; however future efforts can explore using transfer/few-shot learning between different 3D rigid-body systems.

### 7.3 POTENTIAL NEGATIVE SOCIETAL IMPACTS

While we do not believe this work directly facilitates injury to living beings, the inconsistency between the predicted latent representation and ground truth data may lead to unexpected results if deployed in real world environments.

### 7.4 FUTURE WORK

Although our approach so far has been limited to embedding RGB-images of rotating rigid-bodies with configuration spaces in $\mathbf{SO}(3)$, it is important to note that there are natural extensions to a wider variety of problems. For instance, this framework can be extended to embed different high-dimensional sensor measurements—such as point clouds—by only modifying the feature extraction layers of the autoencoder. Likewise, depending on the rigid-body system, the latent space can be chosen to reflect the appropriate configuration space, such as generic rigid-bodies in $\mathbf{SE}(3)$ or systems in more complicated spaces, such as the $n$-jointed robotic arm on a restricted subspace of $\Pi_{i=1}^{n}\left(\mathbf{SO}(3)\right)$.

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

# A APPENDIX

## A.1 AUTO-ENCODER ARCHITECTURE

Tables 2 and 3 give the architecture of the encoder/decoder neural networks. The encoder and decoder combine convolutional and linear layers to map from images to the desired latent space. The nonlinear activation function used in the auto-encoding neural network is the exponential linear unit (ELU). This activation function was chosen for continuity, as well as to prevent vanishing and exploding gradients as shown in Otto and Rowley (2019).

Table 2: Encoder architecture.

| Encoder Layers | | | | |
|---|---|---|---|---|
| Layer Number | Layer Name | Input Channel Size | Output Channel Size | Kernel/Stride Size |
| 1 | Conv2d | 3 | 16 | 3 |
| 2 | ELU | N/A | N/A | N/A |
| 3 | Conv2d | 16 | 16 | 3 |
| 4 | ELU | N/A | N/A | N/A |
| 5 | MaxPOOL | N/A | N/A | 2/2 |
| 6 | BatchNorm2d | 16 | N/A | N/A |
| 7 | Conv2d | 3 | 16 | 3 |
| 8 | ELU | N/A | N/A | N/A |
| 9 | Conv2d | 16 | 16 | 3 |
| 10 | ELU | N/A | N/A | N/A |
| 11 | MaxPOOL | N/A | N/A | 2/2 |
| 12 | BatchNorm2d | 32 | N/A | N/A |
| 13 | Flatten | N/A | N/A | N/A |
| 14 | Linear | 32*4*4 | 120 | N/A |
| 15 | ELU | N/A | N/A | N/A |
| 16 | BatchNorm2d | 120 | N/A | N/A |
| 17 | Linear | 120 | 84 | N/A |
| 18 | ELU | N/A | N/A | N/A |
| 19 | Linear | 84 | 6 | N/A |

Table 3: Decoder architecture.

| Decoder Layers | | | | |
|---|---|---|---|---|
| Layer Number | Layer Name | Input Channel Size | Output Channel Size | Kernel/Stride Size |
| 1 | Linear | 6 | 84 | N/A |
| 2 | ELU | N/A | N/A | N/A |
| 3 | Linear | 84 | 120 | N/A |
| 4 | BatchNorm2d | 120 | N/A | N/A |
| 5 | ELU | N/A | N/A | N/A |
| 6 | Linear | 120 | 32*4*4 | N/A |
| 7 | Unflatten (32, 4, 4) | N/A | N/A | N/A |
| 8 | BatchNorm2d | 32 | N/A | N/A |
| 9 | MaxUNPOOL | N/A | N/A | 2/2 |
| 10 | ELU | N/A | N/A | N/A |
| 11 | ConvTranspose2d | 16 | 16 | 3 |
| 12 | ELU | N/A | N/A | N/A |
| 13 | ConvTranspose2d | 16 | 3 | 3 |
| 14 | BatchNorm2d | 16 | N/A | N/A |
| 15 | MaxUNPOOL | N/A | N/A | 2/2 |
| 16 | ELU | N/A | N/A | N/A |
| 17 | ConvTranspose2d | 16 | 16 | 3 |
| 18 | ELU | N/A | N/A | N/A |
| 19 | ConvTranspose2d | 16 | 3 | 3 |

# B    BASELINES

We compare the performance of our model against three baseline architectures: (1) an LSTM baseline, (2) a Neural ODE (Chen et al., 2018) baseline, and the Hamiltonian Generative Network (HGN) (Toth et al., 2020). The first two architectures use the same encoder-decoder backbone as our model (see Tables 2 and 3) the HGN is trained with the architecture described in their work. The baselines differ from our approach in the way the dynamics are computed.

## B.1    LSTM - BASELINE

The LSTM-baseline uses an LSTM network to predict the dynamics. The LSTM-baseline is a three-layer LSTM network with an input dimension of 6 and a hidden dimension of 50. The hidden state and cell state are randomly initialized and the output of the network is mapped to a 6-dimensional latent vector by a learned linear transformation. We train the LSTM-baseline to predict a single step from 9 sequential latent vectors by minimizing the sum of the autoencoder and dynamics losses, $\mathcal{L}_{ae}$ and $\mathcal{L}_{dyn}$ as defined in equations 4.4.1 and 7, and the MSE loss between the encoder generated latent vectors and the LSTM-baseline latent vectors. The baseline is trained using the hyper-parameters in Tables 5, 7, 8, and 6. At inference we use a recursive strategy to predict farther into the future. The qualitative performance for the LSTM-baseline is given in figure 4, and the quantitative performance in terms of pixel mean-square error given in Table 1. The total number of parameters in the network is 52400.

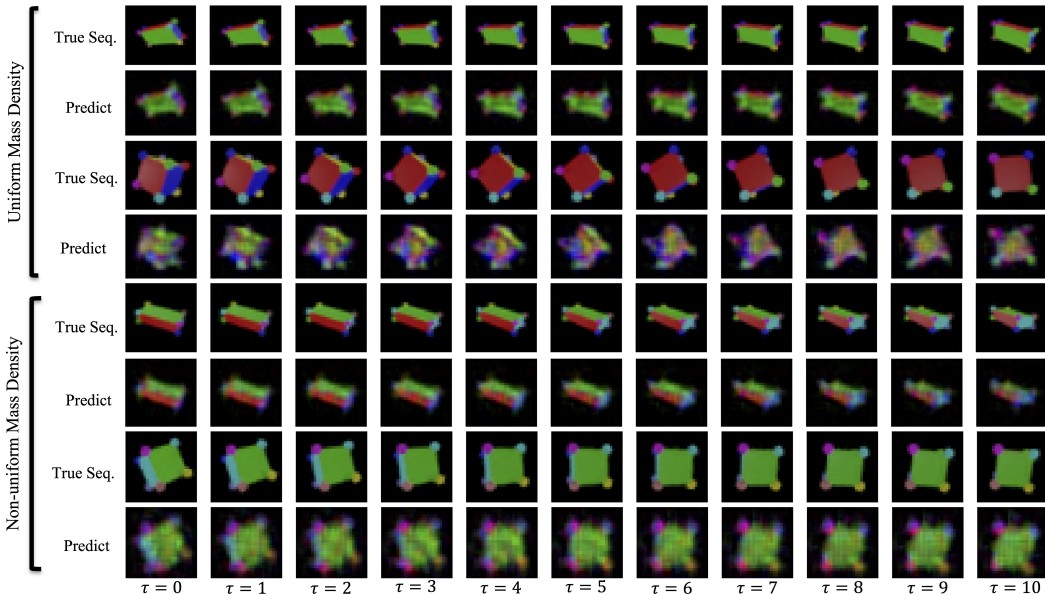

Figure 4: Predicted sequences for uniform/non-uniform prism and cube datasets given by the LSTM-baseline. At prediction time, the model takes the first 9 encoded latent states and predicts a sequence of 20 time steps recursively. The figure shows time steps 10-19, which are the first 10 predictions of the model. The LSTM-baseline has poorer performance than the proposed approach on all datasets.

## B.2    NEURAL ODE (CHEN ET AL., 2018) - BASELINE

The Neural ODE-baseline uses Neural ODE (Chen et al., 2018) framework to predict dynamical updates. The Neural ODE-baseline is a three layer MLP with ELU Clevert et al. (2015) activations. The baseline has an input dimension of 6, a hidden dimension of 50, and an output dimension of 6. We train the Neural ODE-baseline to predict a sequence of latent vectors from an single latent vector input by minimizing the sum of the autoencoder and dynamics losses, $\mathcal{L}_{ae}$ and $\mathcal{L}_{dyn}$ as defined in equations 4.4.1 and 7, and the MSE loss between the encoder generated latent vectors and the Neural ODE-baseline latent vectors. We use the RK4-integrator to integrate the dynamical update and the hyper-parameters in Tables 5, 7, 8, and 6. The qualitative performance for the NeuralODE-baseline is

given in figure 5, and the quantitative performance in terms of pixel mean-square error given in table 1. The total number of parameters in the network is 11406.

Figure 5: Predicted sequences for uniform/non-uniform prism and cube datasets given by the Neural ODE-baseline. At prediction time, the model takes the first encoded latent states and predicts a sequence of 10 steps recursively. The Neural ODE-baseline has poorer performance than the proposed approach on all datasets.

### B.3 HAMILTONIAN GENERATIVE NETWORK (HGN)

HGN (Toth et al., 2020) uses combination of variational auto-encoding neural networks, transformer, and Hamiltonian dynamics to do video prediction. For our comparison experiments, we use the implementation Balsells Rodas et al. (2021). During training, the same training hyperparameters are used as described in Toth et al. (2020). We train HGN on our four datasets using their loss function (i.e. reconstruction and KL divergence). We use the Leap-frog integrator as this provided their model with its best performance on their datasets. The qualitative performance for HGN on our datasets is given in figure **??**, and the quantitative performance in terms of pixel mean-square error given in table 1.

## C  ABLATION STUDY

In this ablative study we explore the impacts of the dynamics loss 7 and energy loss 4.5 on the performance of the proposed model. The ablated model is trained similarly to the proposed model except the dynamics (or energy) loss is not present in the total loss. From table 4 and figure 6 it can be seen that removing the dynamics loss negatively affects the performance of our proposed model. Inspecting table 4 it can be seen that removing energy loss has a minimal degradation effect on model performance. These results are further corroborated in the literature (Allen-Blanchette et al., 2020; Watter and Jost Tobias Springenberg, 2015).

## D  EXPERIMENT HYPER-PARAMETERS

### D.1  UNIFORM MASS DENSITY CUBE

$$\mathbf{J}_{\text{cube}}^{-1} = \begin{bmatrix} 3. & 0. & 0. \\ 0. & 3. & 0. \\ 0. & 0. & 3. \end{bmatrix}$$

Table 4: Average pixel MSE over a 30 step unroll on the train and test data on four datasets for our ablative study. All values are multiplied by 1e+3. We evaluate our model and compare to a version of our model without the dynamics loss 7 and without the energy-base loss. Our full model outperforms the ablated models in the prediction task across all datasets.

| Dataset | Ours | | Ours - dynamics | | Ours - energy | |
|---|---|---|---|---|---|---|
| | TRAIN | TEST | TRAIN | TEST | TRAIN | TEST |
| Uniform Prism | **2.66 ± 0.10** | **2.71 ± 0.08** | 6.41 ± 2.02 | 6.40 ± 1.98 | 3.03 ± 1.26 | 3.05 ± 1.21 |
| Uniform Cube | **3.54 ± 0.17** | **3.97 ± 0.16** | 11.30 ± 1.47 | 11.30 ± 1.51 | 4.13 ± 2.14 | 4.62 ± 2.02 |
| Non-uniform Prism | **4.27 ± 0.18** | **6.61 ± 0.88** | 8.80 ± 2.51 | 8.75 ± 2.51 | 4.98 ± 1.26 | 7.07 ± 1.88 |
| Non-uniform Cube | **6.24 ± 0.29** | **4.85 ± 0.35** | 13.84 ± 1.62 | 13.83 ± 1.66 | 7.27 ± 1.06 | 5.65 ± 1.50 |

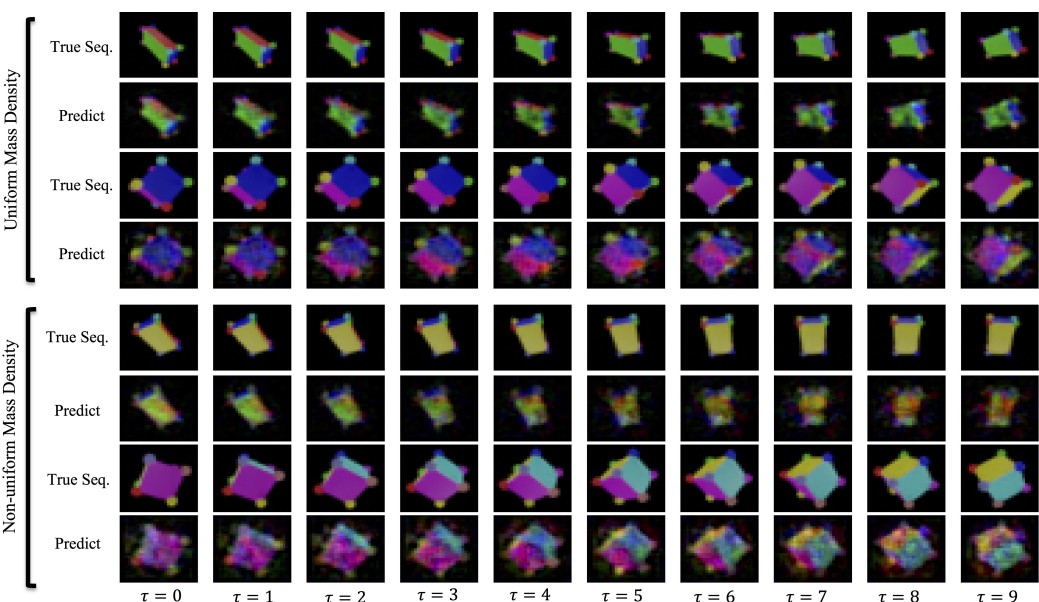

Figure 6: Predicted sequences for uniform/non-uniform prism and cube datasets given by the ablated version of our proposed model. At prediction time, the model takes the first encoded latent states and predicts a sequence of 10 steps recursively. The ablated model has poorer performance than the proposed approach on all datasets.

The training parameters used to run this experiment with the `train_dev.py` file is given in table 5.

Table 5: Hyper-parameters used to train model for the uniform mass density cube experiment. Only values differing from default values are given in the table.

| Uniform Cube Experiment | |
|---|---|
| Parameter Name | Value |
| seed | 17 |
| single-gpu | True |
| test_split | 0.2 |
| val_split | 0.1 |
| n_epoch | 1000 |
| batch_size | 256 |
| learning_rate_ae | 1E-3 |
| learning_rate_dyn | 1E-3 |
| seq_len | 10 |
| time_step | 1E-3 |
| loss_gamma | 1. 1. 1. 0.1 0.1 |

## D.2 UNIFORM MASS DENSITY PRISM

$$\mathbf{J}_{\text{prism}}^{-1} = \begin{bmatrix} 2.4 & 0. & 0. \\ 0. & 0.71 & 0. \\ 0. & 0. & 0.6 \end{bmatrix}$$

The training parameters used to run this experiment with the `train_dev.py` file is given in table 6.

Table 6: Hyper-parameters used to train model for the uniform mass-density prism experiment. Only values differing from default values are given in table.

| Uniform Prism Experiment | |
|---|---|
| Parameter Name | Value |
| seed | 17 |
| single-gpu | True |
| test_split | 0.2 |
| val_split | 0.1 |
| n_epoch | 1000 |
| batch_size | 256 |
| learning_rate_ae | 1E-3 |
| learning_rate_dyn | 1E-3 |
| seq_len | 10 |
| time_step | 1E-3 |
| loss_gamma | 1. 1. 1. 0.1 0.1 |

## D.3 NON-UNIFORM DENSITY CUBE

$$\mathbf{J}_{\text{nu-cube}}^{-1} = \begin{bmatrix} 4.52677669 & -2.61906366 & -0.43651061 \\ -2.61906366 & 1.34388683 & -0.77601886 \\ -0.43651061 & -0.77601886 & -0.12933648 \end{bmatrix}$$

The training parameters used to run this experiment with the `train_dev.py` file is given in table 7.

Table 7: Hyper-parameters used to train model for the non-uniform mass density cube experiment. Only values differing from default values are given in table.

| Non-uniform Cube Experiment | |
|---|---|
| Parameter Name | Value |
| seed | 17 |
| single-gpu | True |
| test_split | 0.2 |
| val_split | 0.1 |
| n_epoch | 1000 |
| batch_size | 256 |
| learning_rate_ae | 1E-3 |
| learning_rate_dyn | 1E-3 |
| seq_len | 10 |
| time_step | 1E-3 |
| loss_gamma | 1. 1. 1. 0.1 0.1 |

## D.4 NON-UNIFORM DENSITY PRISM

$$\mathbf{J}_{\text{nu-prism}}^{-1} = \begin{bmatrix} 2.36999579 & 0.12136536 & 0.39443742 \\ 0.12136536 & 0.67762326 & 0.2022756 \\ 0.39443742 & 0.2022756 & 0.6573957 \end{bmatrix}$$

The training parameters used to run this experiment with the `train_dev.py` file is given in table 8.

Table 8: Hyper-parameters used to train model for the non-uniform mass density prism experiment. Only values differing from default values are given in table.

| Non-uniform Prism Experiment | |
|---|---|
| Parameter Name | Value |
| seed | 17 |
| single-gpu | True |
| test_split | 0.2 |
| val_split | 0.1 |
| n_epoch | 1000 |
| batch_size | 256 |
| learning_rate_ae | 1E-3 |
| learning_rate_dyn | 1E-3 |
| seq_len | 10 |
| time_step | 1E-3 |
| loss_gamma | 1. 1. 1. 0.1 0.1 |

# E  ANGULAR VELOCITY ESTIMATOR

The angular velocity is estimated using two sequential images frames, $I_0$ and $I_1$. The image frames are encoded into two latent states, $\mathbf{R}_0$ and $\mathbf{R}_1$,. These latent states are the orientation matrices in the body-fixed frame. The angular velocity between frame 0 and frame 1 is calculated in two parts: (1) the unit angular velocity vector and (2) the angular velocity magnitude. The unit vector and magnitude are estimated as shown in Figure 1.

# F  COMPUTE RESOURCES & GPUS

The models are trained on a server with 8 NVIDIA A100 SXM4 GPUs. The processor is an AMD EPYC 7763, with 64 cores, 128 threads, 2.45 GHz base, 3.50 GHz boost, 256 MB cache, PCIe 4.0.

# G  CODE REPOSITORY & DATASET

For access to the code repository, go to the Github link: `https://github.com/jjmason687/LearningSO3fromImages`. For access to data used in this work please see link at `https://www.dropbox.com/sh/menv3lu9mquu1wh/AABovQ53udtryDC24xPLGw17a?dl=0`.

