# OpenReview forum: "Learning Interpretable Dynamics from Images of a Freely Rotating 3D Rigid Body"
_ICLR.cc/2023/Conference — Submitted to ICLR 2023_

### Official Review · Reviewer_14VD · 2022-10-21

**Confidence:** 2
**Correctness:** 4
**Technical Novelty And Significance:** 3
**Empirical Novelty And Significance:** 3
**Recommendation:** 5

**Clarity, Quality, Novelty And Reproducibility:**

I think that the writing is sufficiently comprehensible except in few sections. The main concepts are made quite clear. The limitations and failure cases have been discussed and the future works have been correctly identified. The authors will release the code upon acceptance to make the method reproducible. The main contribution is novel in the context of satellite trajectory estimation.

**Strength And Weaknesses:**

Strengths:
1. The use of hamiltonian dynamics is novel to the context of rotation estimation and trajectory prediction of satellites.
2. A dataset has been proposed in order to learn 3D dynamics through images.
3. The experiments show convincing results.

Weakness:
1. No evaluation on real data.
2. In figure 4, it seems that the rotation estimation starts well and drifts away quickly. The rotation trajectory prediction is very off while the image prediction is quite decent. Do the images in fugure 3 (upto t=50) represent a small portion of the trajectory show on figure 4. More details are needed.

**Summary Of The Paper:**

The paper presents a method to estimate 3D rotations of an object from images using classical physics. The paper is focused on predicting  satellite rotations. It uses Hamiltonian representation of rotational motion. With this representation, motion trajectory is differentiable everywhere. This is not true for rotation parametrisation using exponential or quaternion maps as the inverses may be discontinuous. The paper provides a 3D dataset of uniform and non-uniform cube and prism to train the method.
The method uses a set of sequential images to estimate rotation and angular velocities. It encodes images to low-dimensional space representing rotations (learnt during training phase) which are used to estimate angular velocity and predicts next image using Hamiltonian dynamics.

**Summary Of The Review:**

The paper is definitely interesting. My only concern is the lack of evaluation on real data and the inability of method to deal with non-uniform data.

After reading the rebuttal and discussion with other reviewers, I have changed my opinion (a bit unfavorably). Here are the main concerns:

1. The experimentation is highly limited. The discussion on handling non-linear mass distribution is non-existent. The authors provide some explanation in the rebuttal but a lot questions remain. Given that the setup is difficult, perhaps the authors could explore non-linearity a bit more. For example, how does the method behave when the mass is non-uniform in a way that the centre of mass deviates strongly with respect to object centre. This discussion could have helped to understand the practical limitations of the method better. Plus, the authors must experiment with more data. Right now, the datasets are limited to a cube and a prism.

2. There were some gaps in writing. The figure 2 in initial draft, which shows the trajectory to drift apart drastically, was later on removed. It is unclear to the readers how the registration works if the estimation of trajectory is so wrong. It is not clear whether the authors are performing a global alignment to fit into the projection. Perhaps the authors should have clarified.

Overall, the paper has an important contribution over sota but the technical novelty is limited. Without a strong experimental evaluation, it was difficult to have an undoubted positive opinion.

---

> ### Author Response · Authors · 2022-11-19
> **Response to Reviewer 14VD**
>
> We thank the reviewer for taking time to engage with our work. Our response to the comments and specific questions are as follows:
>
> * **No evaluation on real data.**
>     We think evaluation on real data would be interesting and important experiments for future work. However, we note that our scenarios are not simple, notably because while the surface of the rigid body is always regular, the mass distribution is not. Please see the general statement above that we will add to clarify this point. We introduce a new dataset inspired by Homeomorphic Variational Auto-Encoding [4] but with additional complexity. Specifically, our dataset includes both a cube and prism with uniform and non-uniform mass distributions. So, while the external shapes are relatively simple, the internal mass distributions, which matters fundamentally for the dynamics, are not. That our model can differentiate between cases with the same shape but different mass distributions indicates that our model is in fact learning the dynamics independent of the shape. Additionally, we have found that datasets of simple renderings are commonly used in the literature. For example, the related work Hamiltonian Neural Networks [1] reports experimental results on the OpenAI gym [2] pendulum environment with the pendulum angle restricted, and Hamiltonian Generative Networks [3] uses renderings of Gaussian blobs for point mass depictions of various mechanical systems. Note that we introduce our dataset because neither of these datasets are appropriate for our task (i.e., predicting rigid body dynamics).
>
> * **In figure 4, it seems that the rotation estimation starts well and drifts away quickly. The rotation trajectory prediction is very off while the image prediction is quite decent. Do the images in fugure 3 (upto t=50) represent a small portion of the trajectory show on figure 4. More details are needed.**
> We thank the reviewer for catching this. We agree the figure does not show alignment between the ground truth and learned trajectories, nor should it since the learned moment of inertia matrix is not necessarily in the principle axis frame and the ground truth is. We have removed the figure in our revised submission.
>
> [1] Greydanus, Samuel, Misko Dzamba, and Jason Yosinski. "Hamiltonian neural networks." Advances in neural information processing systems 32 (2019).
> [2] Brockman, Greg, et al. "Openai gym." arXiv preprint arXiv:1606.01540 (2016).
> [3] Toth, Peter, et al. "Hamiltonian generative networks." arXiv preprint arXiv:1909.13789 (2019).
> [4] Falorsi, Luca, et al. "Explorations in homeomorphic variational auto-encoding." arXiv preprint arXiv:1807.04689 (2018).

---

> > ### Comment · Area_Chair_fjxR · 2022-11-23
> > **Following up on rebuttal**
> >
> > Dear reviewer,
> >
> > the authors tried to respond to your criticism regarding generalization to real data. Do you find the responses satisfactory, and how does it influence your opinion?
> >
> > Cheers,
> > Your AC

---

> > > ### Comment · Reviewer_14VD · 2022-11-29
> > > **Real data experiments**
> > >
> > > I understand that experimenting on real data might be difficult. But, at this stage, it is not clear how the method can adapt to rigid bodies with significantly different mass distribution , given that the real data is more complex than a cube/prism. I think that the paper needs an crude analysis on weight distribution on satellite data (which is completely missing) or they really need to test with ‘very’ diverse mass distributions.
> > >
> > > My slightly positive opinion about the paper is due to the challenging problem it is solving. Learning dynamics on a non-uniform mass distribution is a step forward in this research. The only missing part is that how much this method can adapt to real conditions.

---

> > > > ### Author Response · Authors · 2022-11-29
> > > > **Response to comment on experiments with different mass distribution**
> > > >
> > > > Thank you for your further comments. We want to highlight that our framework is designed to learn the dynamics of a rigid body with any mass distribution. In general, the mass of a rigid body is not uniformly distributed, e.g., in the case of a satellite. Thus, in general, the mass distribution is independent of the body’s visual appearance. The fact that our framework is designed to handle any non-uniform mass distribution is connected to the fact that the dynamics, as described by Hamilton’s equations on SO(3), of rigid bodies with different non-uniform distributions of mass only differ with respect to the direction of the “principal” axes. This follows because the moment of inertia matrix J is, by definition, always a positive definite matrix, no matter how complicated the distribution of mass. Therefore, J is always diagonalizable – it is diagonal when written with respect to the principal axes. In our paper, we explore cubes and prisms where there is a concentration of mass in one of the corners. This we view as a rather “extreme” case of a non-uniform mass distribution and we show that our model can learn the dynamics. We apologize for not clarifying the nature of the non-uniformity sooner. We will do so in the final version of the paper. Generalizing to cases in which the mass distribution changes in time, i.e., to non-rigid bodies, is not captured by our model and presents an interesting future direction.

---

### Official Review · Reviewer_7a5H · 2022-10-26

**Confidence:** 4
**Correctness:** 3
**Technical Novelty And Significance:** 3
**Empirical Novelty And Significance:** 3
**Recommendation:** 5

**Clarity, Quality, Novelty And Reproducibility:**

Clarity: The introduction section should be improved with details of the proposed method. In Fig. 1, what are (a), (b) and (c) actually not clear.

Quality & Novelty: As the paper acknowledged, the paper only evaluated the proposed method in a rather idealized setting. To conclude the current experimental evaluation and ablation studies are insufficient to justify the paper.


Reproducibility: Currently, more details are needed to achieve reproduction of the proposed method.

**Strength And Weaknesses:**

Strength:

+ A physics-informed deep learning framework for predicting image sequences of 3D rigid-bodies by embedding the images as measurements in the configuration space SO(3) and propagating the Hamiltonian dynamics forward in time

+ A new rotating rigid-body dataset with sequences of rotating cubes and rectangular prisms with uniform and non-uniform density.

Weaknesses:

- As the paper acknowledged, the paper only evaluated the proposed method in a rather idealized setting. To conclude the current experimental evaluation and ablation studies are insufficient to justify the paper.

- The proposed approach currently relies on the ability to train the model for each system being examined.

- The secod paragraph of the Summary section seems to be overclaiming its contributions without sufficient justification.

**Summary Of The Paper:**

The paper presented a physics-informed neural network model to estimate and predict 3D rotational dynamics from image sequences. The target is achieved by using a multi-stage prediction pipeline that maps individual images to a latent representation homeomorphic to SO(3), computes angular velocities from latent pairs, and predicts future latent states using the Hamiltonian equations of motion with a learned representation of the Hamiltonian.

**Summary Of The Review:**

The paper presented a physics-informed neural network model to estimate and predict 3D rotational dynamics from image sequences. However, the current experimental evaluation and ablation studies are insufficient to justify the paper. The novelty/contribution of the proposed method over existing work has not been fully justified.

---

> ### Author Response · Authors · 2022-11-19
> **Response to Reviewer 7a5H**
>
> We thank the reviewer for taking time to engage with our work. Our response to the comments and specific questions are as follows:
>
> * **As the paper acknowledged, the paper only evaluated the proposed method in a rather idealized setting. To conclude the current experimental evaluation and ablation studies are insufficient to justify the paper.**
> While idealized in some respects (e.g., no clutter in the images, no noise in the measurement), we note that our scenarios are not quite as simple as they may appear, notably because while the surface of the rigid body is always regular, the mass distribution is not. We introduce a new dataset inspired by Homeomorphic Variational Auto-Encoding [1] but with additional complexity. Specifically, our dataset includes both a cube and prism with uniform and non-uniform mass distributions. So, while the external shapes are relatively simple, the internal mass distributions, which matters fundamentally for the dynamics. That our model can differentiate between cases with the same shape but different mass distributions indicates that our model is in fact learning the dynamics independent of the shape. Additionally, we have found that datasets of simple renderings are commonly used in the literature. For example, the related work Hamiltonian Neural Networks [4] reports experimental results on the OpenAI gym [5] pendulum environment with the pendulum angle restricted, and Hamiltonian Generative Networks [2,3] uses renderings of Gaussian blobs for point mass depictions of various mechanical systems. Note that we introduce our dataset because neither of these datasets are appropriate for our task (i.e., predicting rigid body dynamics).
> * **The introduction section should be improved with details of the proposed method. In Fig. 1, what are (a), (b) and (c) actually not clear.** We thank the reviewer for their feedback. We have clarified the figure in the revised submission by using (top), (bottom-left) and (bottom-right) instead of (a), (b), and (c).
> * **The proposed approach currently relies on the ability to train the model for each system being examined.**
> We agree with this assessment but find that this is common in the literature (consider, for example [4,2,7,6]).
> * **The second paragraph of the Summary section seems to be overclaiming its contributions without sufficient justification.**
> We thank the reviewer for their feedback. We have removed the last sentence of the second paragraph in the revised submission. We feel the remaining claims, those touting the benefits of interpretable representations in deep neural networks, are substantiated in the literature ([7,8,9]), and the work we have presented.
> * **Clarity: The introduction section should be improved with details of the proposed method…**
> We thank the reviewer for their feedback. We have added additional details to the introduction (please see the general statement above).
> **Reproducibility: Currently, more details are needed to achieve reproduction of the proposed method.**
> We thank the reviewer for their feedback. We would like to point out that we provide the details of our model architecture in Appendix A, our hyperparameters in Appendix D, and we have stated in Appendix G that we plan to make the code repository and image datasets available upon acceptance.
>
> [1] Falorsi, Luca, et al. "Explorations in homeomorphic variational auto-encoding." arXiv preprint arXiv:1807.04689 (2018).
> [2] Toth, Peter, et al. "Hamiltonian generative networks." arXiv preprint arXiv:1909.13789 (2019).
> [3] Irina Higgins, Peter Wirnsberger, Andrew Jaegle, Aleksandar Botev, "SyMetric: Measuring the Quality of Learnt Hamiltonian Dynamics Inferred from Vision," NeurIPS 2021, 2021.
> [4] Greydanus, Samuel, Misko Dzamba, and Jason Yosinski. "Hamiltonian neural networks." Advances in neural information processing systems 32 (2019).
> [5] Brockman, Greg, et al. "Openai gym." arXiv preprint arXiv:1606.01540 (2016).
> [6] Duong, Thai, and Nikolay Atanasov. "Hamiltonian-based neural ODE networks on the SE (3) manifold for dynamics learning and control." arXiv preprint arXiv:2106.12782 (2021).
> [7] Zhong, Yaofeng Desmond, Biswadip Dey, and Amit Chakraborty. "Symplectic ode-net: Learning hamiltonian dynamics with control." arXiv preprint arXiv:1909.12077 (2019).
> [8] Bengio, Yoshua, Aaron Courville, and Pascal Vincent. "Representation learning: A review and new perspectives." IEEE transactions on pattern analysis and machine intelligence 35.8 (2013): 1798-1828.
> [9] Esteves, Carlos, et al. "Learning so (3) equivariant representations with spherical cnns." Proceedings of the European Conference on Computer Vision (ECCV). 2018.

---

### Official Review · Reviewer_RcCn · 2022-10-31

**Confidence:** 4
**Correctness:** 3
**Technical Novelty And Significance:** 3
**Empirical Novelty And Significance:** 2
**Recommendation:** 5

**Clarity, Quality, Novelty And Reproducibility:**

quality

Some variables with hat are not defined.


clarity

Symbols and variables in chapter 3 were defined in chapter 4, which made the paper difficult for me to understand.


originality

Since there are already several methods for learning Hamiltonian systems from video images[1,2], it is difficult to say that the proposed method is novel unless its performance is sufficiently high.

**Strength And Weaknesses:**

Strengths

This research is impressive because it is a challenging task to realize a neural network method that learns a physical model capable of predicting video images. The fact that they have succeeded in predicting and reconstructing video images to some extent by incorporating a physical inductive bias specific to rotational motion into the model is also highly novel.



Weaknesses

The following two points are considered weaknesses


The advantage over previous research is unclear:

Hamiltonian Generative Networks (HGN) [1] has been shown to be able to learn up to 400 steps in the future for a two-body problem [2].
HGN is also more general-purpose than the proposed method.
The authors need to show that the proposed method has better prediction performance for videos than HGN.

[1] Peter Toth, Danilo J. Rezende, Andrew Jaegle, Sébastien Racanière, Aleksandar Botev, Irina Higgins, "Hamiltonian Generative Networks," ICLR 2020, 2019.

[2] Irina Higgins, Peter Wirnsberger, Andrew Jaegle, Aleksandar Botev, "SyMetric: Measuring the Quality of Learnt Hamiltonian Dynamics Inferred from Vision," NeurIPS 2021, 2021.



The purpose of the method is unclear:.

The author writes that the application of the study is to estimate the orientation of satellites, but the training data in this study assigns a different color to each side of the object. I think it does not match the configuration of the satellites. If the differences between each side of a satellite can be determined, the orientation and motion of a satellite with a known geometry can be estimated analytically. Therefore, the proposed method is not considered necessary to achieve the objectives of the study.

Also, the work [Duong and Atanasov (2021)] cited by the authors in section "2.3 LEARNING DYNAMICS FOR RIGID BODIES" to achieve more general learning of rigid body motions by estimating inverse mass properties could be used for satellite orientation estimation, but no comparative experiments have been conducted.

These factors made me think that the relationship between the proposed method and the purpose of the study did not seem appropriate.

**Summary Of The Paper:**

The manuscript proposes a method for estimating and predicting 3D rotation dynamics from image sequences of freely rotating 3D rigid bodies such as satellites. The proposed method maps individual images to latent representations isomorphic to SO(3) using auto encoder type neural networks, calculates angular velocities from the latent pairs, and then leams the moment of inertia tensor J in the Hamiltonian. Based on this prediction of future latent states, a predictive image is constructed using the auto encoder. After constructing a new rotating rigid body dataset consisting of a sequence of rotating cubes and rectangles, the authors show that the proposed method gives higher performance than LSTM and NeuralODE.

**Summary Of The Review:**

The importance of the research is high because it is a challenging task to realize a neural network method that learns a physical model capable of predicting video images. However, the performance of the method has not been compared to similar more general-purpose methods, which makes its effectiveness questionable. Therefore, I believe this concern needs to be resolved to accept this manuscript.

---

> ### Author Response · Authors · 2022-11-19
> **Response to Reviewer RcCn**
>
> We thank the reviewer for taking time to engage with our work. Our response to the comments and specific questions are as follows:
>
> * **The authors need to show that the proposed method has better prediction performance for videos than HGN**
> Per the reviewer’s feedback, we have applied HGN to our rotating rigid body dataset using the pytorch implementation[7]. In summary, we find that HGN is unable to capture the dynamics as well as our model. Quantitatively, the mean squared error is higher than for our model (see Table 1). Moreover, since the latent state is represented by a vector in $\mathbb{R}^n$, and the moment of inertia matrix isn’t explicitly modeled, it may be more difficult to integrate into downstream tasks such as control. We will include qualitative results in the camera ready version of the paper.
>
> * **The author writes that the application of the study is to estimate the orientation of satellites, but the training data in this study assigns a different color to each side of the object. I think it does not match the configuration of the satellites…**  Indeed, our approach is motivated by application to learning the dynamics of satellites as well as other bodies in space. We agree that the different colors do not necessarily match the configuration of a satellite. However, our objective here is to learn the dynamics of the rigid body by observing its motion and not its external geometry, since the external geometry does not inform the dynamics. Please see the general statement above that we have added to the paper. We also note that datasets of simple renderings are commonly used in the literature. For example, the related work Hamiltonian Neural Networks [4] reports experimental results on the OpenAI gym [5] pendulum environment with the pendulum angle restricted, and Hamiltonian Generative Networks [2] uses renderings of Gaussian blobs for point mass depictions of various mechanical systems. Note that we introduce our dataset because neither of these datasets are appropriate for our task (i.e., predicting rigid body dynamics).
>
> * **If the differences between each side of a satellite can be determined, the orientation and motion of a satellite with a known geometry can be estimated analytically.**
> This is not true, except in the very special case in which the mass of the rigid body is known and the mass distribution is known to be uniformly distributed over the volume of the body. Please see the general statement above, which we will add to the paper to clear up any confusion on the principles of rigid body rotational dynamics. Importantly, the dynamics of a rigid body, namely how the angular velocity of the rigid body evolves over time, are not a function of the geometry of the body but rather the distribution of mass, which when not uniformly distributed leads to rotational motion that is very different from that of a body with uniform density. So knowing the geometry (in this case the color of the faces of the satellite) is not enough to predict dynamics in general.
>
> * **Also, the work [Duong and Atanasov (2021)] … could be used for satellite orientation estimation, but no comparative experiments have been conducted.**
> We thank the reviewer for their feedback, however, we do not see how [6] is comparable to our approach. The authors in [6] propose a method to learn the inverse moment of inertia from low-dimensional measurement data, and we propose a method to simultaneously learn an embedding to a low-dimensional state space and the moment of inertia from image data. We consider this to be a major contribution since the naive extension of [6] to the image setting (adding an encoding and decoding network and optimizing over the reconstruction error) performs poorly in practice. Moreover, we consider our approach to be well motivated since in many real-world settings image observations may be available when low-dimensional measurement data are not.
>
> * **Some variables with hat are not defined.** We thank the reviewer for catching this. We have added the following statement about hatted variables to Section 4.1 Notation: “Quantities generated with the learned dynamics are denoted with a hat (e.g., $\hat{q}$).”
>
> * **Symbols and variables in chapter 3 were defined in chapter 4, which made the paper difficult for me to understand.** We thank the reviewer for catching this. We have resolved this in the revised submission.

---

> > ### Author Response · Authors · 2022-11-19
> > **References for our response**
> >
> > [1] Falorsi, Luca, et al. "Explorations in homeomorphic variational auto-encoding." arXiv preprint arXiv:1807.04689 (2018).
> > [2] Toth, Peter, et al. "Hamiltonian generative networks." arXiv preprint arXiv:1909.13789 (2019).
> > [3] Irina Higgins, Peter Wirnsberger, Andrew Jaegle, Aleksandar Botev, "SyMetric: Measuring the Quality of Learnt Hamiltonian Dynamics Inferred from Vision," NeurIPS 2021, 2021.
> > [4] Greydanus, Samuel, Misko Dzamba, and Jason Yosinski. "Hamiltonian neural networks." Advances in neural information processing systems 32 (2019).
> > [5] Brockman, Greg, et al. "Openai gym." arXiv preprint arXiv:1606.01540 (2016).
> > [6] Duong, Thai, and Nikolay Atanasov. "Hamiltonian-based neural ODE networks on the SE (3) manifold for dynamics learning and control." arXiv preprint arXiv:2106.12782 (2021).
> > [7] Rodas, Carles Balsells, Oleguer Canal, and Federico Taschin. "Re-Hamiltonian Generative Networks." ML Reproducibility Challenge 2020. 2021.

---

> > > ### Comment · Reviewer_RcCn · 2022-12-13
> > > **Revise of my rating**
> > >
> > > My rating was revised because the author's response to another reviewer's point about the lack of practical experimental seemed inadequate.

---

### Official Review · Reviewer_togy · 2022-10-31

**Confidence:** 3
**Correctness:** 4
**Technical Novelty And Significance:** 3
**Empirical Novelty And Significance:** 3
**Recommendation:** 8

**Clarity, Quality, Novelty And Reproducibility:**

The background section is very concise but sufficiently clear for readers that are familiar with Hamiltonian formulation of rigid bodies dynamics.
As claimed by the authors, and to the best of my knowledge it is the first physics-informed deep learning algorithm for predicting image sequences of 3D rigid bodies.

A few questions:

How figure 4 should be interpreted? I struggle to understand why the predicted $S^{2}\times S^{2}$ parameterisation of SO(3) is so different from the ground truth. In this setting there is a lot of prior knowledge used in the algorithm, namely we know that the system is a freely rotating rigid body. Therefore, as you emphasise, "learning dynamics" boils down to making good predictions. In figure 3 I observe seemingly good predictions that I can't really conciliate with figure 4, where trajectories clearly differ. Section 6.2 does not provide sufficient explanation for why this is the case.

Because of the freedom os assigning an inertial frame Eq. (4) (and the corresponding Lie-Poisson) and (2) are independent. It is not very clear why you need the latter system of equations? Technically you could perform predictions using only the quaternion equations. Is it only because you want to enforce energy conservation? If so it would be interesting to know how the algorithm performs without this penalty in the loss.

**Strength And Weaknesses:**

The methods makes good use of well-known classical mechanics results about freely rotating rigid bodies giving good theoretical reasons for why the method should work. Interpretability of the learned dynamics is of utmost importance when using machine learning to model dynamical systems and the proposed algorithm gives very interpretable results constraining the dynamics on SO(3) and leaving all the unknown and irrelevant characteristics, such as color and shape of the modelled object, to the autoencoder.

Unfortunately, as stated in the paper, there aren't many datasets to assess performances for the particular problem tackled in this paper.
The dataset could also represent a good benchmark for future works if made available, and therefore is a notable contribution.
The comparison with LSTMs and Neural ODEs is really meaningful as these methods are not as domain-specific as the proposed algorithms but certainly show how inductive biases like the one proposed in this paper are the way to go. Is there any

Apart from the dynamics, the SO(3) latent space seems to facilitate the reconstruction: perhaps a pre-training of just the encoder-decoder architecture, which is not unrealistic for real-world application, would boost performances. Or perhaps you already do that.



**Summary Of The Paper:**

The paper propose a method to learn dynamics and make predictions from sequences of images of 3D rigid bodies. The method incorporates physics priors by using SO(3) as the latent space, and by translating the problem of learning the dynamics to that of learning a (reduced) Hamiltonian. Predictions are then made by and integrating the kinematic equations on SO(3), and Lie-Poisson equations on the Lie coalgebra of SO(3).

They perform experiments with a synthetic dataset consisting of sequences of rotating rigid bodies with different shapes and different, diagonal and non-diagonal, inertia tensors.

**Summary Of The Review:**

The paper propose a method to learn dynamics of freely rotating rigid bodies based on the simple idea that in the Hamiltonian framework, rigid bodies dynamics happens on $T^{*}SO(3)$. This makes learned dynamics interpretable and is a concrete step towards more trustworthy, less black-box machine learning models. Despite I asked for a few clarifications, I think the paper is overall clear.

---

> ### Author Response · Authors · 2022-11-19
> **Response to Reviewer togy**
>
> We thank the reviewer for taking time to engage with our work. Our response to the comments and specific questions are as follows:
>
> * **The dataset could also represent a good benchmark for future works if made available, and therefore is a notable contribution.** We thank the reviewer for their suggestion. We will make the dataset available upon acceptance.
>
> * **[Would] pre-training of just the encoder-decoder architecture … boost performances**
> We think this could be an interesting direction to investigate further. However, we consider the ability to train our model end-to-end, with a single objective function, a strength of our approach.
>
> * **How should figure 4 be interpreted? I struggle to understand why the predicted S^2 X S^2 parameterisation of SO(3) is so different from the ground truth…**
> We thank the reviewer for catching this. We agree the figure does not show alignment between the ground truth and learned trajectories, nor should it since the learned moment of inertia matrix is not necessarily in the principle axis frame and the ground truth is. We have removed the figure in our revised submission.
>
> * **Technically you could perform the prediction using only the quaternions...**
> This is not true, except in the very special case in which the mass of the rigid body is known and the mass distribution is known to be uniformly distributed over the volume of the body. Please see the general statement above, which we will add to the paper to clear up any confusion on the principles of rigid body rotational dynamics. Importantly, the dynamics of a rigid body, namely how the angular velocity of the rigid body evolves over time, are not a function of the geometry of the body but rather the distribution of mass, which when not uniformly distributed leads to rotational motion that is very different from that of a body with uniform density. So knowing the geometry (in this case the color of the faces of the satellite) is not enough to predict dynamics in general.
>
> * **... it would be interesting to know how the algorithm performs without [the energy conservation] penalty in the loss.**
> Per the reviewer’s suggestion, we have performed an ablation study for the energy-based loss function. The ablation results are included in Appendix C of our revised submission. In summary, we find that removing the energy-based loss minimally impacts quantitative performance. In retrospect this is unsurprising, since our dynamical equations are designed to respect conservation of energy.

---

### Author Response · Authors · 2022-11-19
**General Statement**

We’ve noticed some confusion amongst the reviewers regarding the importance of dynamics in our framework and thus the significance of our results. It seems that more than one reviewer has suggested that with sufficient knowledge of external geometry of the rigid body, the latent trajectories can be predicted using only the kinematics. This is not true since the motion of a rotating rigid body depends fundamentally on its moment of inertia matrix, which is a function of its mass and mass distribution. It is only in the special case in which the mass of the rigid body is known and known to be uniformly distributed over the volume that the moment of inertia matrix can be calculated based on the external geometry of the body. We aren’t assuming a known mass. Furthermore, in real world problems, like the ones that motivate this work, rigid bodies won’t have uniformly distributed mass. Thus, predicting the motion of a rigid body requires learning its rotational dynamics. We note that in the general case of a rigid body with nonuniform density those dynamics will typically be very different from the rotational dynamics of a rigid body with the same geometry but with uniform density. A major result of our paper is an architecture that learns (from images) the rotational dynamics in the general case, where learning the shape of the body is insufficient to predict rigid body rotational motion. To address this confusion, we provide the following clarification statement which will be added to our paper.

Kinematics and dynamics of 3D rigid body rotation are both fundamental to accomplishing the goals of this paper. The kinematics describe the rate of change of rigid body orientation as a function of the orientation and the angular velocity. Our method integrates the kinematic equations to compute the orientation trajectory in latent space using the latent angular velocity. The dynamics describe the rate of change of the angular velocity as a function of the angular velocity and the moment-of-inertia matrix J, which depends on the distribution of mass over the rigid body volume. J is unknown and cannot be computed from knowledge of the external geometry of the rigid body, except in the special case in which the mass is known and the mass is uniformly distributed over the rigid body volume. In our framework, we learn the dynamics from the motion of the rigid body, not from the external geometry since the special case is not practical. Importantly, the difference between the dynamics of a uniformly distributed mass and a non-uniformly distributed mass inside the same external geometry is significant. We show that we can learn these very different dynamics even when the external geometry is the same. By integrating the learned dynamics, we can predict future latent angular velocity. Works such as [1,2] estimate the transformation between image pairs, which can then be used to generate image trajectories. However, this approach implicitly assumes the object evolves with a constant velocity, an assumption that is not true in the general case. Our work addresses this limitation by estimating the angular acceleration and mass distribution for the object in addition to its angular velocity.

[1] Falorsi, Luca, et al. "Explorations in homeomorphic variational auto-encoding." arXiv preprint arXiv:1807.04689 (2018).

[2] Connor, M. C. and Rozell, C. J. “Representing Closed Transformation Paths in Encoded Network Latent Space” arxiv:1912.02644(2019)

---

### Decision · Program_Chairs · 2023-01-20

**Decision:**

Reject

**Justification For Why Not Higher Score:**

The empirical findings must become more convincing and generalizable

**Justification For Why Not Lower Score:**

The key idea is great.

**Metareview: Summary, Strengths And Weaknesses:**

This paper presents a very interesting idea, which conceptually I am very much in favor of. Specifically, the authors propose to learn dynamics from pixels (not given perfect state information), by positing as a learning problem in a latent manner. Particularly, they propose latent representations that are homeomorphic to SO(3).

The idea is great, however, there was significant scepticism about the results and the complexity of the experiments. A danger in this line of works is having published works that emphasize a lot on the theoretical aspects, however, they do not sufficiently quantify the success in practice. Now, this was a sufficiently hard case, just because the idea is so nice. At the end of the day, however, the experiments were on arguably data settings that are visually at least underwhelming. Having a rotating cube centered on the frame, with highly distinct colors does not make much of a case for learning dynamics directly from pixels, in other words the paper in its current form overclaims.

**Summary Of Ac-Reviewer Meeting:**

Indeed, this was certainly a borderline paper. We had an AC-reviewer meeting, in which we all agreed that the idea is very interesting, however the results are not as persuasive as one would expect for ICLR. Indeed, having non-homogeneous densities is interesting, however, this does not make up for a visually harder setting, "only" the dynamics become more complex (by "only" here, I do not mean that this is a simple problem. What i mean is that then the problem is not as much on how to learn from pixels, which is the key interesting aspect of this work, but how to learn more complex dynamics from a simple visual setting).

I believe this work has great potential, and highly encourage the authors to strengthen it further empirically, either with more complex backgrounds (see variants of CLEVR or the object-centric learning literature)., or visually more complex systems (eg pendula).